# A Prospective Observational Cohort Study on Pharmacological Habitus, Headache-Related Disability and Psychological Profile in Patients with Chronic Migraine Undergoing OnabotulinumtoxinA Prophylactic Treatment

**DOI:** 10.3390/toxins11090504

**Published:** 2019-08-29

**Authors:** Marialuisa Gandolfi, Valeria Donisi, Fabio Marchioretto, Simone Battista, Nicola Smania, Lidia Del Piccolo

**Affiliations:** 1Department of Neurosciences, Biomedicine and Movement Sciences, University of Verona, 37134 Verona, Italy; 2UOC Neurorehabilitation, AOUI Verona, 37134 Verona, Italy; 3Section of Clinical Psychology, Department of Neurosciences, Biomedicine and Movement Sciences, University of Verona, 37134 Verona, Italy; 4Unit of Neurology, IRCCS Sacro Cuore Don Calabria Hospital, 37024 Negrar (Verona), Italy

**Keywords:** psychological distress, pain, disability, Anti-Inflammatory Agents Non-Steroidal, drugs

## Abstract

Chronic Migraine (CM) is a disabling neurologic condition with a severe impact on functioning and quality of life. Successful therapeutic management of patients with CM is complex, and differences in therapeutic response could be attributable to genetically determined factors, sensitivity to pharmacological treatment, psychosocial and relational factors affecting the patient’s compliance and approach on the therapeutic treatment. The aim of this prospective observational study was to explore self-efficacy, coping strategies, psychological distress and headache-related disability in a cohort of 40 patients with CM (mean age: 46.73; standard deviation 13.75) treated with OnabotulinumtoxinA and the relationship between these clinical and psychological aspects and acute medication consumption during OnabotulinumtoxinA prophylactic treatment. Patients presented an overall significant reduction in the Headache Index (HI) (*p* < 0.001), HI with severe intensity (*p* = 0.009), and total analgesic consumption (*p* = 0.003) after the prophylactic treatment. These results are in line with the literature. Despite this, higher nonsteroidal anti-inflammatory drugs consumption was associated with higher psychological distress, higher HI with severe and moderate intensity, and worse quality of life. Conversely, triptans consumption was correlated with HI of mild intensity, and problem-focused coping strategies. To conclude, the psychological profile, and in particular, the psychological distress and specific coping strategies might influence the self-management of acute medication.

## 1. Introduction

Chronic Migraine (CM) is a disabling neurologic condition with a severe impact on functioning and quality of life [1]. It accounts for about one-third of chronic headaches affecting 1–2% of the general population [1]. According to the Classification ICHD-III (beta), patients with CM experience headache at least 15 days per month for more than three consecutive months, with at least eight days per month being classified as migraine attacks or headaches that respond to migraine-specific medications [2]. This definition has two main implications: (i) it takes into account the presence of migraine attacks of different severity (i.e., lighter attacks) without neurogenerative hallmarks that can resemble tension-type headache; (ii) it introduces new diagnostic criteria which also include patients with overuse of acute migraine medications [1]. 

The physiopathology of CM is not clearly understood [1,3]. However, multifactorial pathophysiology in which susceptibility to attack-triggering stimuli contribute to the threshold for the generation of migraine attack has been suggested [1]. As far as the neurogenic inflammation of meninges is considered, high levels of cytokines and neuropeptides are released during migraine attacks. They activate specific neural pathways that transmit pain to the trigeminovascular and vegetative system nuclei. The release of these neuropeptides induces vasodilatation in meningeal areas and neurogenic inflammation. Repeated episodes of trigeminovascular system activation sensitize peripheral nociceptors and central pain pathways leading to central sensitization [4]. Central and peripheral nociceptors sensitization is assumed to be one of the key mechanisms of migraine chronification [1]. Increased migraine attack frequency along with insufficient acute pain relief lead to sensitization, which further lower migraine attack threshold [1].

With this perspective, migraine is conceptualized as a continuum from episodic to chronic forms, whereby “chronic” indicates severity or patient-specific symptomatic burden, as opposed to the duration of disease [5]. On the one hand, many contributing factors can decrease the sensory threshold for the generation of migraine attacks (i.e., genetic predisposition, female sex, obesity, depression, and stressful life events). An ineffective acute treatment and overuse of acute migraine medication predispose to migraine chronification. Conversely, factors related to behaviour and lifestyle (i.e., higher education, being married, social support, exercise and stress management) have been reported as protective factors that can heighten the threshold and thereby reduce the frequency of migraine attacks. The evidence that about 26% of patients with CM go into remission within two years of chronification emphasized the importance of early and multidisciplinary management [1]. 

Successful management of patients with CM is complex and represents a challenge for primary care as well as for specialist, as in other chronic pain conditions [3,6]. Particularly pertinent for the management of CM is the systematic monitoring of migraine symptoms using the headache diary to have an accurate reporting of headache frequency and intensity [6]. From a clinical point of view, this approach allows estimating the Headache Index (HI) as a measure of disease severity and impact in the quality of life. In the research field, having a comparable index of disease severity permits to compare treatment effects among studies.

It has been well established that the treatment of acute attacks and prophylactic management play a crucial role. However, each subject does not have the same therapeutic response [3]. The reason for these different outcomes should be searched for in different domains that could be attributable to genetically determined factors interfering with the aetiology, tendency to the chronification, sensitivity to pharmacological treatment, psychosocial and relational factors affecting the patients’ compliance and approach to the treatment [1,3].

As far as prophylactic management, OnabotulinumtoxinA is the only prophylactic treatment specifically approved to reduce the frequency and the severity of migraine attacks [7,8,9]. In 2010, it was approved for the treatment of chronic migraine after two large-scale double-blind, randomized placebo-controlled trials which supported its safety and effectiveness [7,8,9]. Recent guidelines recommended OnabotulinumtoxinA as an effective and well-tolerated treatment of CM and provided recommendations for its use [10]. However, future studies on the treatment of CM using OnabotulinumtoxinA are needed to improve the management of this disabling disorder [10]. The mechanism of action of OnabotulinumtoxinA in central neuropathic pain has been described in detail in the literature [11]. The potential action on sensitized nociceptive afferents is intriguing. In rats, OnabotulinumtoxinA was shown to decrease sensitization on nociceptive meningeal C-fibers [12]. The hypothesis is that OnabotulinumtoxinA can directly decrease the amount of calcitonin gene-related peptide released from trigeminal neurons [13] and thus reduce neurogenic inflammation and central sensitization [13]. More recently, novel therapeutic options aimed at inhibiting calcitonin gene-related peptide signaling has attracted growing interest in the pharmacological research on migraine. Human monoclonal antibodies have been approved for the preventive treatment in patients with episodic and chronic migraine, reporting positive effects [14,15,16].

Nevertheless, positive results should rely on the management by the patient of acute medication. For example, the treatment of acute attacks is prescribed by the specialist but self-managed by the patient according to the intensity and the features of each migraine attack. They include the use of different categories of drugs, such as nonsteroidal anti-inflammatory drugs (NSAIDs) and triptans, usually in polytherapy. The prophylactic treatment through OnabotulinumtoxinA reduces the headache symptoms and common comorbidities, including depression and anxiety [17,18,19,20,21,22]. Briefly, these studies reported that OnabotulinumtoxinA alleviated the severity and frequency of migraine attacks and these effects were associated with improvements in depression and anxiety. These improvements might be a consequence of the CM relief. However, the molecular mechanisms are unknown.

Note that psychological factors and pain coping styles might influence acute medication intake in terms of quantity and type (i.e., NSAIDs and triptans) [17,23]. Literature has suggested, for instance, a relationship between migraine episode recurrence and maladaptive coping strategies [24]. Gunel and Akkaya reported that patients with migraine tend to seek social support lesser than healthy controls [25]. Bagianti et al. (2014) reported in their retrospective analysis on patients with chronic migraine and medication overuse that at one-year follow-up patients who did relapse into medication overuse reported increased anxiety and perpetuation of severe dependency-like behaviours. Conversely, patients who did not relapse had clinical improvements with decreased depressive symptoms and dependency-like behaviour, although reporting a low internal pain control [26]. More recently, D’Amico et al. (2015) found that disability and quality of life were moderately or little related to clinical and psychological factors in patients with CM and medication overuse [27].

The main limitations of these studies are the lack of homogeneity about the prophylactic treatment and, where it was provided, considerable heterogeneity in the OnabotulinumtoxinA injection paradigm (and dose). Moreover, the lack of assessment of acute medication consumption, and the evaluation of mainly depression and anxiety should be outlined.

So far, no studies exploring the influence of psychological distress and coping strategies on the OnabotulinumtoxinA treatment effects and the acute medication consumption in patients with CM have been reported. 

This study aimed to explore self-efficacy, coping strategies, psychological distress and headache-related disability in a cohort of patients with CM treated with OnabotulinumtoxinA and the relationship between these clinical and psychological aspects and acute medication consumption during OnabotulinumtoxinA prophylactic treatment.

We expect that the relative contribution of psychological factors would vary among patients undergoing the same prophylactic treatment and would interfere with the acute medication consumption. Improved understanding of the association between these factors might lead us to more effective management for this disabling neurologic condition.

## 2. Results

### 2.1. Demographic and Clinical Carachteristics of Patients with CM

Seventy-two patients attending the Neurorehabilitation Unit were screened for eligibility. Eighteen patients were excluded because involved in other research activities. Fifty-four patients were considered eligible. Fourteen patients were excluded because of the lack of consent to perform the psychological assessment and incomplete diary in the 12 weeks before the enrolment. Therefore, 40 patients with CM (mean age: 46.73 years; SD: 13.75) were enrolled.

The majority of patients was women (92.5%), most of whom were contraceptive consumers (89.2%) and menopausal (59.5%). Most patients were both smoker (85%) and alcohol consumers (87.5%); a large percentage of them was already under pre-study Headache Prophylaxis therapy (60%) and did not respond to acute headache medication (65%). The majority of patients were not taking antidepressants or anxiolytic therapy (65%). Patients’ demographic and clinical features are reported in detail in Table 1.

### 2.2. Headache Index (HI) and Headache-Related Disability Scores during Treatment Phases

There was an overall significant reduction in the HI (*p* < 0.001). Similarly, the HI with severe intensity (*p* = 0.009), and total analgesic consumption (*p* = 0.003) decreased significantly over time. Post hoc analysis resulted in a significant reduction in the HI (T1: *p* < 0.001; T2: *p* = 0.008), in the HI with severe intensity at T2 (*p* = 0.021) and in total analgesic consumption at T2 (*p* = 0.017). HI with moderate and mild intensity progressively decreased, albeit not significantly.

Quality of life (HIT 6) showed an improvement over time, albeit not statistically significant. At T0, the high HIT-6 mean score (above 60 points) demonstrated the severe impact of CM symptoms on quality of life. At T2, the HIT-6 score decreased by −2.95 points indicating a meaningful improvement in the quality of life (within-person MCI: −2.5 points) and a substantial reduction of symptoms severity. The NSAIDs and Triptans consumption progressively decreased over time, albeit not significantly. Headache Index and Analgesic Daily Consumption are reported in Table 2. 

### 2.3. Psychological Characteristics of Patients with CM

Self-efficacy, Coping Strategies and Disability are reported in Table 3. As regards coping strategies (COPE) the mean score of the sample indicates that the majority of patients included in our study were comparable to those reported in the Italian validation study of COPE scale [28]. The mean value of the self-efficacy (GSE) scale is in line with the previous value reported by D’Amico and collaborators in a sample of chronic migraine patients with medication overuse [27]. Conversely, the average psychological distress global score (GSI) was higher than 0.57, which has been considered a cut-off value for psychological dysfunction [29,30]. The majority of patients (85%) reported a CM related severe disability at T0, as measured by the Migraine Disability Assessment (MIDAS).

### 2.4. Correlation between Clinical Outcomes and Psychological Profile

There was a positive correlation between the headache-related disability and the total analgesic consumption (r_s_ = 0.336, *p* =0.034), the HI (r_s_ = 0.547, *p* = 0.003), the HI with severe intensity (r_s_ = 0.379, *p* = 0.024) and quality of life (r_s_ = 0.464, *p* =0.003). 

In general, a higher analgesic consumption was associated with higher HI (r_s_ = 0.547, *p* < 0.001). However, NSAID consumption was positively correlated with higher HI (r_s_ = 0.489, *p* = 0.001) and with severe intensity (r_s_ = 0.343, *p* = 0.037), while triptans consumption was correlated with HI of mild intensity (r_s_ = −0.390, *p* = 0.017).

At T0, there was a positive correlation between psychological distress (GSI) and NSAIDs consumption, HI with severe intensity and headache-related disability (HIT-6). Higher positive attitude (COPE-PA) correlated with higher analgesic consumption. Similarly, there was a positive correlation between problem-focused strategies (COPE-PO) and the total analgesic and triptans consumption. Higher perceived social support (COPE-SS) positively correlated with HI with mild intensity (Table 4 and Figure 1).

Changes in pharmacological intake showed that higher psychological distress (GSI) correlated with higher triptans consumption at T1 and T2. When considering the GSI cut-off (0.57), patients without psychological dysfunction (GSI values < 0.57) reported a reduction in the triptans consumption while patients with psychological disfunction (GSI ≥ 0.57) increased the triptans consumption (Figure 1).

Avoidance strategies (COPE-AS) were positively associated with higher total analgesic consumption and higher triptans consumption at T1 and T2. Conversely, higher problem-focused approach (COPE-PO) related to a reduction in the total analgesic and NSAIDs consumption at T2. Turning to religion (COPE-SS) was positively associated with total analgesic consumption (Table 5).

## 3. Discussion

To the best of our knowledge, this study is the first one exploring the pharmacological habitus and the clinical and psychological aspects of CM undergoing the same prophylactic treatment by using OnabotulinumtoxinA. Strengths in the methodology of this study include the fact that all patients underwent the same CM management, consisting of the same specialist giving the diagnosis and pharmacological indication and the same specialist giving the OnabotulinumtoxinA injection according to international guidelines protocol [10].

The main finding of this prospective observational study was that despite the global reduction in the headache symptoms and acute medication intake, higher NSAIDs consumption was associated with higher psychological distress, higher HI with severe intensity, and worse quality of life. Conversely, triptans consumption was correlated with HI of mild intensity, and high problem-focused coping strategies. These two different patterns of medication use suggest that psychological distress and specific coping strategies might influence the self-management of acute medication. 

Additionally, we explored the relationship between psychological aspects and clinical and medication outcomes at follow up. With progress on the prophylactic treatment, pharmacological treatment remained related to the psychological profile. In particular, the reduction in the triptans consumption was related to lower psychological distress and lower avoidance coping strategies. Conversely, lower problem-focused coping strategies were related to an increase in NSAIDs and analgesic consumption, but only at T2. The overall trend in the reduction of the number of days with migraine was similar to that reported in OnabotulinumtoxinA trails. The OnabotulinumtoxinA was first used in the treatment of movement disorders and focal spasticity in patients with Central Nervous System damage. Early observations in cervical dystonia suggested that besides the anticholinergic effect, OnabotulinumtoxinA may decrease pain before muscle relaxation [12]. A report of pain relief in patients with migraine pain, and without muscle hyperactivity, paved the way to the potential OnabotulinumtoxinA mechanism in reducing chronic pain syndromes. Finally, two large-scale double-blind, randomized placebo-controlled trials supported its safety and effectiveness in patients with CM [7,8,9]. Briefly, the standardized PREEMPT (Phase III Research Evaluating Migraine Prophylaxis Therapy) injection protocol consists of OnabotulinumtoxinA injection at a minimum dose of 155U OnabotulinumtoxinA into seven specific muscle groups (fixed-dose paradigm): Frontalis, Corrugator, Procerus, Temporalis, Cervical Paraspinal Muscle group, Occipitalis, Trapezius. A total dose up to 195U was approved for additional injections into the temporalis, occipitalis and trapezius muscles according to a “follow-the-pain” paradigm [31,32,33]. Results from the PREEMPT trials have established that OnabotulinumtoxinA is effective in controlling the pain in patients with CM with or without acute medication overuse up to 56 weeks [7,8,9]. More recently, the long-term phase IV study (COMPEL) has reported the efficacy and tolerability of OnabotulinumtoxinA over two years, in which patients received up to nine treatment cycles. These findings have been confirmed by several real-world clinical practice studies from Europe, including the prospective multinational REPOSE and CM-PASS studies [34,35].

The percentage of reduction in the HI at T1 (−16%) and T2 (−18.8%) could be considered beyond the responder rate reported in the literature. It might depend on the fact that patients were undergoing prophylactic treatment using OnabotulinumtoxinA since at least three injections, and thus, they were not naïve patients to the treatment. Therefore, treatment effects should be considered as additional effects obtained after prolonged OnabotulinumtoxinA injection. Interestingly, the percentage of reduction in the total number of analgesic consumptions was around 19% at T1 and approaching 42% at T2. It might depend on how the headache diary was compiled: data were self-reported and could be influenced by the patient’s mood, with the risk of overvaluing headache symptoms. Conversely, the number of medication intake should be considered a more objective measure of therapy efficacy.

Patients reported severe adverse impact on the quality of life and disability related to CM and, notwithstanding the definite reduction of HIT6 observed over time, the adverse effect was still relevant. Moreover, although examining the prevalence of psychological disorders in the sample was out of the scope of this study, it is noteworthy that approximately a third of patients declared the use of antidepressant or anxiolytic medications and the mean GSI score suggested a high level of psychological distress at enrolment. This result seems in line with the literature, which highlights a high prevalence of psychological disorders, particularly depression and anxiety, and even sub-threshold psychological symptoms in patients with CM [1,27,36,37,38,39,40,41]. The disability picture emerging from those data confirms the relevance of assessing psychological distress, psychiatric comorbidities and quality of life in patients with CM when attending healthcare services for receiving prophylactic treatment using OnabotulinumtoxinA. Indeed, psychological distress is often under-recognized and untreated, and patients with migraine often do not report psychological distress or psychiatric symptoms in the clinical encounter [42]. The continuity of OnabotulinumtoxinA treatment administration overtime for several months might contribute to easing the assessment of psychological distress of patients and referring them, if needed, to psychological treatment. This longitudinal approach should be introduced as a future direction of our study in order to explore the effect of OnabotulinumtoxinA on psychopathology course both alone and combining botulinum and effective psychological approaches [43,44,45].

Higher psychological distress resulted associated both with higher headache-related negative impact and HI severity, confirming the findings reported by previous literature on the relationship between comorbid depression and anxiety symptoms with adverse migraine outcomes [22]. Additionally, even if psychological distress resulted higher in patients with medication overuse, our findings show a specific association with NSAIDs consumptions at baseline, while patients with higher psychological distress reported higher triptans consumptions over time. To the best of our knowledge, this specific relationship between NSAIDs or triptans consumption and psychological distress has not been studied yet. However, preliminary studies in stressed rats with depression and pseudodementia like symptoms showed that the administration of triptans, in particular, zolmitriptan, might act as a 5-HT-1B agonist and therefore decrease depression symptoms [46].

Our sample reported lower perceived self-efficacy than Italian validation study data, according to results reported by D’Amico et al. (2015) [27]. Self-efficacy is conceived as the belief that one can successfully engage in actions to achieve the desired outcome [47,48]. This psychological process has been recognized to interact with headache treatment outcomes. Higher levels of self-efficacy are associated with greater adherence to preventive and acute pharmacological treatments [48]. Conversely, low self-efficacy is a prognostic indicator for negative outcome in chronic headache [49]. In our study, perceived self-efficacy did not significantly correlate with medication use, HI and HIT-6 scores both at baseline and with progress on the prophylactic treatment. Albeit not significant from a statistical point of view, patients with NSAIDs consumption reported a negative correlation with the perceived self-efficacy as if they felt to be less able to manage acute attacks. In the literature, self-efficacy has been examined in a paucity of studies on CM patients; however, this psychological dimension has been suggested as critical for headache management and disability [27,48,50]. Indeed, the defective patients’ confidence in their ability to achieve personal goals along with depressive symptoms might affect the success of therapeutic strategies to manage their acute medication intake. Therefore, self-efficacy might represent a psychosocial aspect of interest, which should be explored in more extensive studies on OnabotulinumtoxinA treatment in CM [51].

Looking at previous literature, Wieser et al. (2011) reported that dysfunctional coping, characterized by fear and avoidance, is frequently used by headache patients [52]. Only a third of all patients in this cross-sectional study on pain coping behavior in patients with migraine and tension-type headache used positive coping strategies with flexible reactions to pain [52]. Russo et al. (2019), in their cross-sectional study, recruited sixty-one patients with migraine without aura who completed three self-reported questionnaires assessing coping strategies [24]. Results suggested that migraine patients showed a low tendency to “turning to religion” approach than healthy controls; however, they did not differ on other coping strategies than healthy controls. Indeed, they did not perceive daily living as more stressful than healthy controls. Due to the sample characteristics and study design, it is not possible to compare our data with those reported by Russo et al. (2019). Our study confirms that coping strategies used by patients with CM are comparable to the general population, observing a slightly higher use of social support and lower use of problem-oriented coping [28]. Interestingly, findings suggest that specific coping strategies might be related to different acute medication management (i.e., NSAIDs and triptans). Note that patients were free for choice in the acute medications use. Moreover, they were invited to respond to self-administered questionnaires, without any specific instruction on coping strategies or self-efficacy; these concepts were described in terms of usual behaviour in dealing with problems or pain to which patients had to respond by choosing the answer that better fitted for them. Our data showed that the overall analgesic consumption at baseline was positively correlated with a positive attitude and problem-solving oriented attitude. However, when analgesic consumption decreased at T2, higher problem-solving coping strategies turned out to be significant only for NSAIDs consumption. In contrast, the increase of analgesic consumption at T1 and specifically, the increase of triptans consumption both at T1 and T2 was related to higher avoidance coping strategies. Considering that the individual attempt to use cognitive and behavioural strategies to manage demands or stress (coping) has been poorly investigated in CM population attending prophylactic treatment and the complexity of CM, the current explorative results should be confirmed in a more extensive study.

To sum up, our findings support the need for patients with CM for a management approach following the bio-psycho-social model. For an appropriate clinical practice during the OnabotulinumtoxinA prophylactic treatment, we recommend an integrated approach, including clinical psychologists working in collaboration with the Neurorehabilitation professionals, to assess psychological needs and profiles. The observed associations between coping strategies, CM disability and pharmacological management highlight the importance of introducing and tailoring psychological interventions considering different psychological factors, including coping strategies. Therefore, the current findings, although being explorative, reinforce the need for using a multidisciplinary approach to implement tailored brief psychological interventions aiming at reducing psychological distress and improving functional coping strategies and self-efficacy in the CM management. As regards the psychological treatment in the neurorehabilitation setting, different approaches, such as cognitive-behavioural, mindfulness, relaxation and biofeedback approaches, have been studied and reviewed resulting valid and safe interventions; and they should be introduced and used in the management of patients with headache [45]. Although these approaches vary somewhat, they have all shown to influence maladaptive behaviours and ways of thinking that contribute to headache-related burden and pain [43,44].

The limitations of our study are the lack of i) follow-up assessments about the psychological outcomes, ii) a control group drug naïve for OnabotulinumtoxinA prophylactic treatment, iii) data on the pre-prophylactic psychological profile. Additionally, the sample was small, and results were based mainly on correlations, which allow only a preliminary exploration of data. Further studies on larger samples of patients should consider these limits and introduce the evaluation of all the clinical and psychological data along with different treatment phases. Furthermore, our sample was mainly composed of females, although this is in line with the significant prevalence of CM among women. Finally, both clinical and psychological data were self-reported.

Future studies should investigate the impact of the prophylactic treatment (OnabotulinumtoxinA or monoclonal antibodies) on the overall status of the patient, to repeat psychological assessment after the first three OnabotulinumtoxinA administration treatment cycles and to examine the combined effects of prophylactic treatment and psychological approaches (e.g., cognitive-behavioural therapy).

## 4. Conclusions

Our study supports that migraine has a multifaceted pathophysiology and suggests that specific sub-population of patients with CM undergoing OnabotulinumtoxinA prophylaxis might take advantage of specific psychological assessment to evaluate psychological profiles relevant to the management of disease. The applicability of our observations in clinical practice requires a multidisciplinary context with the strict collaboration of specialist in neurology, clinical psychology and neurorehabilitation.

## 5. Materials and Methods 

### 5.1. Study Design and Setting

A prospective observational cohort study was performed from March 2018 to February 2019. The study is part of to the project “EXploring PsychoLOgical needs of patients with chronic pain attending neuroREhabilitation services” (EXPLORE) involving the Neurorehabilitation Unit and the Clinical Psychology Unit of the Verona University Hospital and the Neurology Unit of the Sacro Cuore Hospital in Negrar (north Verona). Outpatients with a confirmed diagnosis of CM made by a neurologist according to the Classification ICHD-III (beta) and eligible to OnabotulinumtoxinA prophylaxis were recruited [2,10]. OnabotulinumtoxinA injections were performed at the Neurorehabilitation Unit (AOUI Verona). After the visit, the patients have been informed about the EXPLORE project and asked to participate. After signing written informed consent, included patients filled out psychological questionnaires. Conversely, demographic and clinical data were collected from the medical charts according to clinical practice. The study design is presented in Figure 2.

### 5.2. Participants

Inclusion criteria were: age ≥ 18 years, diagnosis of CM according to the Classification ICHD-III (beta) [2]; ineffective assumption of at least three different drug classes recommended by the International Guidelines of Migraine treatment [10]; undergoing the prophylactic treatment with OnabotulinumtoxinA because intolerance to/inefficacy of primary prophylaxis treatment; diagnosis of other primary headaches.

Exclusion criteria were contraindications to prophylactic treatment using OnabotulinumtoxinA; the presence of other neurologic disturbances that can cause migraine; incomplete diary compiling in the 12 weeks before the enrolment; lack of consent to perform the psychological assessment.

The EXPLORE project was performed according to the latest version of the Declaration of Helsinki and approved by the Ethics Committee of the AOUI of Verona (approval date: 17 January 2018, 1630CESC, Prog. n. 14112).

### 5.3. Measures

#### 5.3.1. Clinical and Demographic Characteristics

Patients recorded headache characteristics using consecutive monthly headache diaries referring in the past three months before the recruitment in the study (T0), when the psychological assessment was performed and continued throughout the study period (T1, T2).

From the headache diaries the following outcomes were recorded: the Headache Index (HI) calculated as the number of headache/migraine days over one month of observation; the intensity of pain calculated as the number of headache/migraine days with mild, moderate and severe pain over the period of observation; the total analgesic consumption (AC) calculated as the number of total analgesics taken during one-month period; the consumption of NSAIDs (NSAIDs C) and triptans (Triptans C) calculated as the number of NSAID and triptans taken in the reference period, respectively.

#### 5.3.2. Headache-Related Disability

At every injection session (T0, T1, T2), the headache-related disability was assessed using the Headache Impact Test (HIT-6) [53,54,55]. The HIT-6 is a six-item validated questionnaire measuring the headache severity and its adverse impact on social functioning, role functioning, vitality, cognitive functioning and psychological distress (higher = worse performance; 60 = severe impact; 56–59 substantial impact; 50–55 some impact; ≤ 49 little to no impact). The within-person Minimally Important Change (MIC) was estimated to be between −2.5 points (mean change approach) and −6 points (Receiver Operating Characteristic - ROC curve approach) [54].

At T0, the Migraine Disability Assessment (MIDAS) Questionnaire was administered to assess the disability induced by the headache during the previous three months. It is a brief, self-administered questionnaire including five items which examine the disability in three fields of daily-life: either school or work, household work, and either family or social or leisure activities. Two additional questions concern: i) the number of days with headache; ii) the pain of headache attacks (on a 0–10 scale) [56].

#### 5.3.3. Psychological Outcomes

The psychological assessment in the EXPLORE project was performed at T0 during the same visit to the OnabotulinumtoxinA treatment session and included a battery of questionnaires. The psychological tools considered in this study are: the General Self efficacy (GSE) scale; the Coping Orientation to Problems Experienced (COPE); the Symptom Checklist-90 (SCL-90-R) questionnaire of which the global severity index (GSI) is reported (average of all SCL-90-R responses) [47,57]. 

The General Self Efficacy Scale (GSE) is a measure of people’s “optimistic self-beliefs to cope with a variety of difficult demands in life”. It consists of 10 items rated on a scale of 1 (not at all true) to 4 (exactly true). The total range between 10 and 40, with higher scores, correspond to greater levels of perceived self-efficacy [47,58].

The new Italian version of the COPE was used to measure strategies by individuals to cope with problems and stress [28]. It is a 60-item self-report measure exploring the following independent coping strategies: positive attitude, social support, problem-solving, avoidance strategies, and turning to religion. Each item is answered on a four-point Likert-type scale ranging from “I usually do not do this at all” to “I usually do this a lot.” 

### 5.4. OnabotulinumtoxinA Treatment

All patients were undergoing OnabotulinumtoxinA prophylaxis according to the international guidelines based on the PREEMPT injection protocol. A total of 195 U was injected every 12 weeks into 39 injection sites (each injection was 5U/0.1 mL) [8,10]. The same expert clinician (MG) performed the treatments as for the fixed-dose paradigm consisting of 155U administered to 31 injection sites across seven head and neck muscles. Besides, 40 U OnabotulinumtoxinA was administered to 8 additional injection sites across three head and neck muscles using a “follow-the-pain” approach.

### 5.5. Study Size

So far, literature has investigated mainly mood and anxiety disorders in patients with CM under Onabotulinumtoxin prophylaxis. Due to the explorative nature of the EXPLORE project and to the lack of studies that have evaluated a broader range of psychological aspects, including self-efficacy perception and coping strategies, it was not possible to calculate the sample size. Therefore, we decided to recruit 40 patients based on the recruitment capability over 11 months at the Neurorehabilitation Unit in the Verona University Hospital.

### 5.6. Statistical Analysis

Descriptive statistics included means, standard deviation and percentage calculation. The Shapiro–Wilk test was used to test data distribution. Non-parametric tests were used for inferential statistics. The Friedman test was used to test for repeated measures differences. Post hoc analysis with Wilcoxon signed-rank tests was conducted with a Bonferroni correction applied, resulting in a significance level set at *p* ≤ 0.025. A Spearman’s rank-order correlation was run to determine the relationship between the psychological outcomes and acute medication consumption and HI. For this purpose, the changes of the score (Δ) between T1-T0 and T2-T0 were computed in the analgesic consumption, NSAIDs consumption and triptans consumption. Higher values corresponded to increased medication consumption at T1 and T2 concerning T0. The level of significance was set *p* ≤ 0.05. Software statistics SPSS 20.0 (IBM SPSS Statistics, Version 22.0, 2013, Armonk, NY, USA).

## Figures and Tables

**Figure 1 toxins-11-00504-f001:**
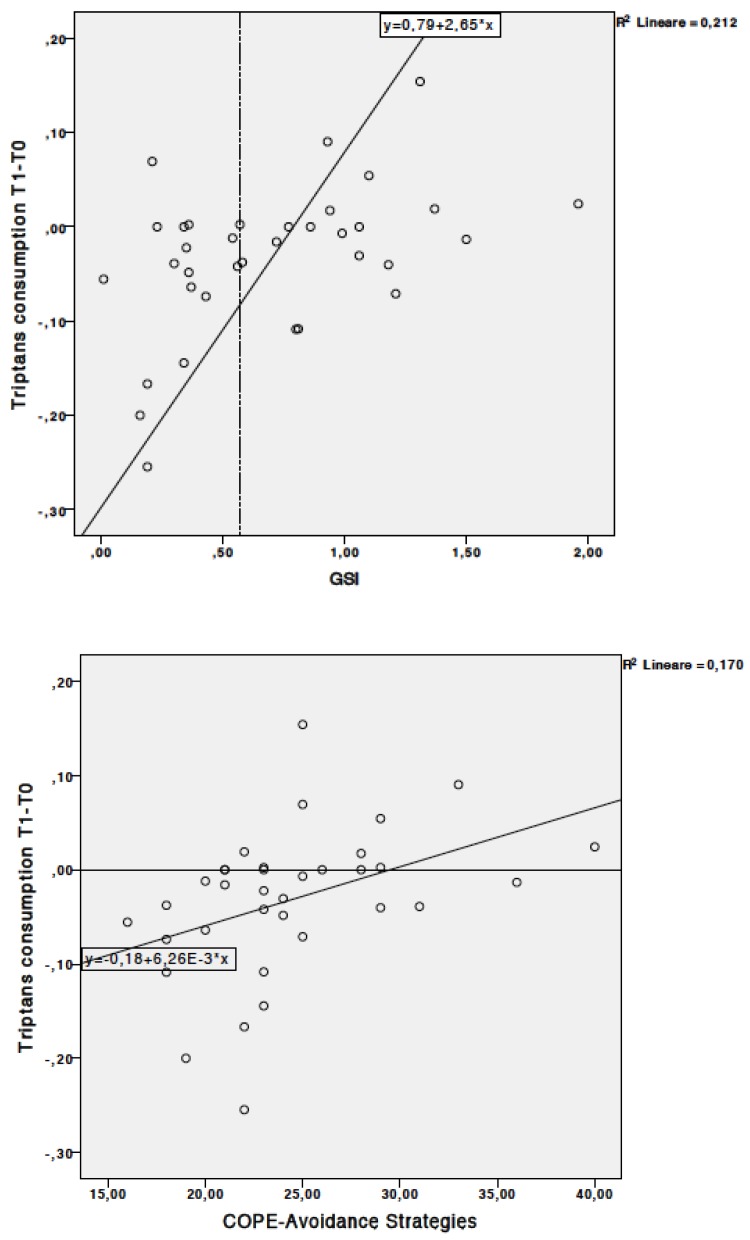
Changes in triptans consumption and psychological distress and avoiding strategies. Panels A (upper): Abscissa indicates the change in triptans consumption between T1 and T0. Ordinate indicates psychological distress (GSI). Panel B (lower): Abscissa indicates the change in triptans consumption between T1 and T0. Ordinate indicates avoidance strategies.

**Figure 2 toxins-11-00504-f002:**
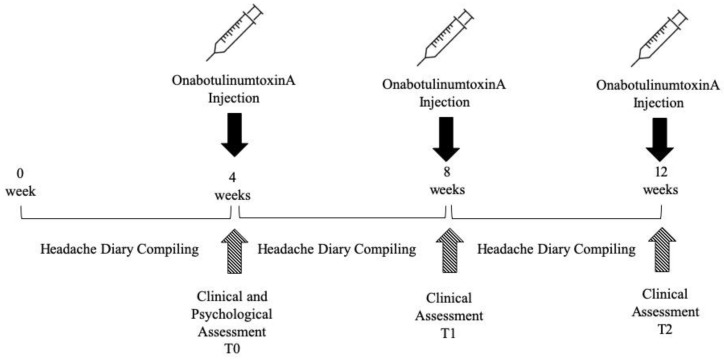
Study design. Patients underwent OnabotulinumtoxinA injection according to the PREEMPT protocol every 12 weeks. The headache diary was completed monthly during the study period. Clinical assessments were performed at T0, T1 and T2. Psychological assessment was performed at T0.

**Table 1 toxins-11-00504-t001:** Patients’ demographic and clinical features.

Demographic and Clinical Features	N (%)
Age (years) (Mean, SD)	46.73 (13.75)
Sex (female)	37 (92.5%)
Body Mass Index	23.62
Sleep regular irregular regular with drugs	17 (42.5%) 15 (37.5%) 8 (20%)
Familiarity with the headache (yes)	25 (62.5%)
Onset pre-adolescence adolescence adult age	11 (27.5%) 13 (32.5%) 16 (40%)
Type of Pain constrictive pulsating both	17 (42,5%) 17(42.5%) 4 (10%)
Presence of aura (yes)	10 (25%)
Pre-study Headache Prophylaxis therapy (non-use)	16 (40%)
Sport Activity (yes)	18 (45%)
Smoker (yes)	34 (85%)
Alcohol consumer (yes)	35 (87.5%)
Contraceptive consumer (yes)	33 (89.2%)
Menopause (yes)	22 (59.5%)
Medical comorbidities (present)	24 (60%)
Acute Headache medicine responsiveness (yes)	14 (35%)
Antidepressants or Anxiolytics consumer (yes)	14 (35%)
On a controlled diet	7 (17.5%)
Number of Detoxification Treatments None One two	19 (47.5%) 18 (45%) 3 (7.5%)

Legend: N, number; %, percentage; SD, standard deviation.

**Table 2 toxins-11-00504-t002:** Headache Index and Analgesic Daily Consumption.

Outcomes	T0 Mean (SD) *n* = 40	T1 Mean (SD) *n* = 35	T2 Mean (SD) *n* = 30	Friedman Test *p*-Value
	Wilcoxon Test for Paired Data			
HI	0.44 (0.25)	0.37 (0.24) **	0.36 (0.24) **	<0.001
HI Severe Intensity	0.92 (1.41)	0.59 (1.09)	0.56 (1.11) *	0.009
HI Moderate Intensity	0.98 (1.39)	0.95 (1.48)	0.78 (1.05)	0.20
HI Mild intensity	1.22 (2.39)	1.36 (2.08)	0.97 (2.20)	0.75
HIT-6	61.62 (8.42)	61.48 (7.57)	58.67 (10.35)	0.06
Total Analgesics Consumption	0.41 (0.45)	0.33 (0.35)	0.24 (0.21) **	0.003
NSAIDs Consumption	0.22 (0.39)	0.17 (0.35)	0.11 (0.21)	0.09
Triptans Consumption	0.18 (0.17)	0.16 (0.17)	0.12 (0.11)	0.10

Legend: SD, standard deviation; HI, Headache Index; HIT-6, Headache Impact Test; *n*, number; NSAIDs, nonsteroidal anti-inflammatory drugs; ** *p* ≤ 0.01 Wilcoxon Test for paired data versus T0; * *p* ≤ 0.05 Wilcoxon Test for paired data versus T0.

**Table 3 toxins-11-00504-t003:** Psychological distress, Self-efficacy, Coping Strategies and Disability.

Outcomes	T0 * Mean (SD)
General Symptoms Intensity (GSI)	0.71 (0.48)
General self-efficacy scale (GSE)	28.45 (4.9)
Coping style: Social support (COPE-SS)	29.98 (6.77)
Coping style: Avoidance Strategies (COPE-AS)	23.97 (5.15)
Coping style: Positive Attitude (COPE-PA)	29.65 (6.30)
Coping style: Problem Oriented (COPE-PO)	30.60 (5.72)
Coping style: Turning to Religion (COPE-TR)	22.07 (5.01)
MIDAS total score	65.67 (61.12)
MIDAS item C headache frequency	39.37 (24.91)
MIDAS item D pain intensity	6.37 (1.61)
MIDAS Disability Categories	*n*, %
Low to Moderate Disability (score < 21)	8 (20)
Severe Disability (score ≥ 21)	32 (80)

Legend: *, *n* = 40; SD, standard deviation; *n*, number; %, percentage; GSI, General Symptoms Intensity; GSE, Perceived Self-efficacy; COPE, coping style; SS, Social support; AS, Avoidance Strategies; PA, Positive attitude; PO, Problem Oriented; TR, turning to religion; MIDAS, Migraine Disability Assessment.

**Table 4 toxins-11-00504-t004:** Spearman correlation between psychological variables and clinical profile at T0.

Outcomes	Total AC	NSAIDs C	Triptans C	HI	HI Severe	HI Mod	HI Mild	HIT-6
*GSI*	0.262 (0.10)	0.410 (0.009) **	−0.151 (0.351)	0.23 (0.159)	0.438 (0.007) **	0.170 (0.315)	0.027 (0.875)	0.394 (0.012) *
*GSE*	−0.09 (0.581)	−0.245 (0.128)	0.111 (0.494)	0.072 (0.661)	−0.170 (0.313)	0.028 (0.87)	0.037 (0.826)	−0.118 (0.469)
*COPE-SS*	0.03 (0.856)	−0.106 (0.515)	0.107 (0.509)	−0.175 (0.279)	0.187 (0.268)	−0.246 (0.142)	−0.43 (0.008) **	0.049 (0.764)
*COPE-AS*	−0.027 (0.871)	0.244 (0.129)	−0.189 (0.242)	0.240 (0.135)	0.257 (0.124)	0.042 (0.804)	0.020 (0.908)	0.174 (0.282)
*COPE-PA*	0.356 (0.024) *	0.008 (0.961)	−0.260 (0.105)	0.092 (0.572)	−0.230 (0.127)	−0.166 (0.325)	−0.244 (0.146)	−0.041 (0.80)
*COPE-PO*	0.354 (0.025) *	−0.04 (0.807)	0.329 (0.038) *	−0.018 (0.913)	−0.257 (0.125)	−0.283 (0.09)	−0.306 (0.066)	−0.287 (0.073)
*COPE-TR*	−0.130 (0.422)	−0.173 (0.285)	0.181 (0.265)	0.109 (0.504)	0.076 (0.657)	0.022 (0.899)	−0.219 (0.193)	0.183 (0.258)

Legend: A, analgesic; C, consumption; HI, Headache Index; Mod, moderate; HIT-6, Headache Impact Test; GSI, General Symptoms Intensity; GSE, Perceived Self-efficacy; COPE, coping style; SS, Social support; AS, Avoidance Strategies; PA, Positive attitude; PO, Problem oriented; TR, turning to religion. Results are displayed as r_s_ (*p* value); r_s_, Spearman’s correlation coefficient; *, significant at *p* ≤ 0.05; **, significant at *p* ≤ 0.01.

**Table 5 toxins-11-00504-t005:** Spearman correlation between psychological variables and change in the acute medication consumption and headache indices during the study.

Outcomes	Total AC T1-T0	Total AC T2-T0	NSAID C T1-T0	NSAID C T2-T0	Triptans C T1-T0	Triptans C T2-T0	HI T1-T0	HI T2-T0
GSI	0.204 (0.239)	0.198 (0.303)	−0.042 (0.811)	−0.030 (0.878)	0.448 ** (0.007)	0.515 ** (0.004)	0.039 (0.826)	−0.032 (0.869)
GSE	−0.191 (0.271)	−0.287 (0.131)	−0.287 (0.094)	−0.157 (0.417)	−0.089 (0.609)	−0.328 (0.083)	0.070 (0.690)	−0.016 (0.936)
COPE-SS	−0.058 (0.740)	−0.074 (0.701)	0.250 (0.148)	0.006 (0.974)	−0.031 (0.862)	−0.015 (0.939)	0.033 (0.849)	−0.081 (0.675)
COPE-AS	0.347 * (0.041)	0.343 (0.069)	−0.064 (0.717)	0.021 (0.913)	0.519 ** (0.001)	0.576 ** (0.001)	0.204 (0.241)	0.100 (0.605)
COPE-PA	−0.075 (0.669)	−0.128 (0.510)	−0.051 (0.771)	−0.107 (0.579)	−0.018 (0.917)	−0.041 (0.833)	−0.104 (0.553)	−0.242 (0.205)
COPE-PO	−0.288 (0.094)	−0.553 ** (0.002)	−0.096 (0.583)	−0.387 * (0.038)	−0.089 (0.612)	−0.295 (0.120)	0.036 (0.839)	−0.102 (0.599)
COPE-TR	0.241 (0.162)	0.441 * (0.017)	−0.170 (0.328)	0.280 (0.141)	−0.035 (0.841)	0.202 (0.294)	−0.227 (0.190)	0.263 (0.168)

Legend: A, analgesic; C, consumption; HI, Headache Index; Mod, moderate; HIT-6, Headache Impact Test; GSI, General Symptoms Intensity; GSE, Perceived Self-efficacy; COPE, coping style; SS, Social support; AS, Avoidance Strategies; PA, Positive attitude; PO, Problem oriented; TR, turning to religion. T0, T1 and T2 are displayed in Figure 2. Results are displayed as r_s_ (*p* value); r_s_, Spearman’s correlation coefficient; *, significant at *p* ≤ 0.05; **, significant at *p* ≤ 0.01.

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
