# Peer review of "A Prospective Observational Cohort Study on Pharmacological Habitus, Headache-Related Disability and Psychological Profile in Patients with Chronic Migraine Undergoing OnabotulinumtoxinA Prophylactic Treatment"

_toxins, 2019, doi:10.3390/toxins11090504_

Round 1

Reviewer 1 Report

This study has for the first time explored self-efficacy, coping strategies, psychological distress and headache-related disability in a cohort of patients with CM treated with BoNT/A and the relationship between these clinical and psychological aspects and acute medication consumption. The research question seems logic to identify these points and their relationships. Literature on these are lacking or limited due to limitation of study design (not including points that are included here) or heterogeneity of populations studied. Hence, this study provides novel and original data. Study however carries some limitations, such as small sample size, nature of having self-reported data, and lack of follow up, that the authors have also recognized nd addressed those at the end of the manuscript. However, the study now has opened the way for further investigation in larger cohorts, and hence, it has a value for presentation.

There are a couple of minor points that authors are encouraged to address in the revised version:

Define HI earlier nd its value of use in migraine related studies. Make it clear as if the participants have had aura or not. Line 16 abstract is difficult to follow, in one hand, it says according to literature and on the other hand the expectation is that the authors present their own data. Perhaps the best is to state this in a way to say data presented that xxxx Add year as unit of age in the abstract. Line 61, after the reference bracket, full stop is missing. Please add to close the sentence. Line 63, the reference style is different, and instead of number, like elsewhere, it is name and year, please correct. In table 2, the stars showing significant difference are presented on other columns than the P value columns, is that right? please double check. The layout of table 3 is not optimal, SD is in a line alone in the middle of the table that can be corrected by correction of the table layout. In another case, (SD, has presented without closing part of the bracket, please make it as (SD) Accordingly, table 4a, and Table 4b layouts are also not optimal, perhaps reduction of font size or change in layout makes it easier to follow lines and numbers presented in these tables. In the discussion, please add whether the participants were given any guides on coping strategies or that option was completely free for choice. What are those strategies and which ones are the most used ones? What are the recommendations of authors for clinicians based on their findings from this study? It is valuable to add few main points to the end of the discussion on applicability of their findings. The paper will benefit from a proof-read for the language edits.

Author Response

Manuscript ID: toxins-576518

Title: A prospective observational cohort study on pharmacological habitus, headache-related disability and psychological profile in patients with chronic migraine undergoing OnabotulinumtoxinA prophylactic treatment.

Dear Editor,

I wish to thank you for your assistance with the manuscript entitled “A prospective observational cohort study on pharmacological habitus, headache-related disability and psychological profile in patients with chronic migraine undergoing OnabotulinumtoxinA prophylactic treatment.”

The manuscript has been revised according to all the Reviewers concerns. The introduction and discussion have been modified to reply to the Reviewers’ suggestions. The reference section has been updated accordingly. Tables have been improved in readability. The language has been reviewed to correct any mistake and typos.

Below a point-by-point reply to the Reviewers' comments is reported.

Thank you, once again, for considering Toxins as a potential home for our paper.

Yours sincerely,

Marialuisa Gandolfi, MD, PhD

Reviewers' comments:

Reviewer #1: This study has for the first time explored self-efficacy, coping strategies, psychological distress and headache-related disability in a cohort of patients with CM treated with BoNT/A and the relationship between these clinical and psychological aspects and acute medication consumption. The research question seems logic to identify these points and their relationships. Literature on these are lacking or limited due to limitation of study design (not including points that are included here) or heterogeneity of populations studied. Hence, this study provides novel and original data. Study however carries some limitations, such as small sample size, nature of having self-reported data, and lack of follow up, that the authors have also recognized and addressed those at the end of the manuscript. However, the study now has opened the way for further investigation in larger cohorts, and hence, it has a value for presentation.

There are a couple of minor points that authors are encouraged to address in the revised version:

Define HI earlier and its value of use in migraine related studies.

Authors reply

In the introduction we define the value of use HI in migraine management and related studies. We clarified that “Successful management of patients with CM is complex and represents a challenge for primary care as well as for specialist, as in other chronic pain conditions [3,6]. Particularly pertinent for the management of CM is the systematic monitoring of migraine symptoms using the Headache diary to have an accurate reporting of headache frequency and intensity [6]. From a clinical point of view, this approach allows estimating the Headache Index (HI) as a measure of disease severity and impact in the quality of life. In the research field, having a comparable index of disease severity permits to compare treatment effects among studies.”

Make it clear as if the participants have had aura or not.

Authors reply

The percentage of patients with aura has been added in table 1.

Line 16 abstract is difficult to follow, in one hand, it says according to literature and on the other hand the expectation is that the authors present their own data.

Authors reply

We rephrased the sentence at line 16 as follows: “Data presented an overall significant reduction in the Headache Index (HI) (p<0.001), HI with severe intensity (p=0.009), and total analgesic consumption (p=0.003) was reported after prophylactic treatment. These results are in line with the literature.”

Perhaps the best is to state this in a way to say data presented that xxxx. Add year as unit of age in the abstract. Line 61, after the reference bracket, full stop is missing. Please add to close the sentence. Line 63, the reference style is different, and instead of number, like elsewhere, it is name and year, please correct.

Authors reply

We add year as unit of age in the abstract. Line 61, after the reference bracket, full stop was added to close the sentence. Line 63, the reference style has been corrected

In table 2, the stars showing significant difference are presented on other columns than the P value columns, is that right? please double check. The layout of table 3 is not optimal, SD is in a line alone in the middle of the table that can be corrected by correction of the table layout. In another case, (SD, has presented without closing part of the bracket, please make it as (SD). Accordingly, table 4a, and Table 4b layouts are also not optimal, perhaps reduction of font size or change in layout makes it easier to follow lines and numbers presented in these tables.

Authors reply

Table 2 has been improved in readability. The p-value in the last column refers to the first level of analysis (Friedman test p-value), while the stars refer to the Wilcoxon test for paired data T1 versus T0 and T2 versus T0. We clarified this point, adding a new row at the top of the table. Table 3 layout has been improved by including column labels in the same column. Typos have been corrected. Table 4a and 4b layout has been changed to present numbers in the same rows.

In the discussion, please add whether the participants were given any guides on coping strategies or that option was completely free for choice. What are those strategies and which ones are the most used ones? What are the recommendations of authors for clinicians based on their findings from this study? It is valuable to add few main points to the end of the discussion on applicability of their findings.

Authors reply

In the discussion section, we clarified that patients were free for choice in the use of acute medications and that participants were invited to respond to self-administered questionnaires, without any specific instruction on coping strategies or self-efficacy; these concepts were described in terms of usual behavior in dealing with problems or pain to which patients had to respond by choosing the answer that better fitted for them. As to the most used strategies we included the study by Wieser et al. (2011) reporting that dysfunctional coping, characterized by fear and avoidance is frequently used by headache patients. Only a third of all patients in his cross-sectional study on pain coping behavior in patients with migraine and tension type headache used positive coping strategies with flexible reactions to pain. Finally, we included our suggestions for clinicians based on our findings as follows: “To sum up, our findings support the need for patients with CM for a management approach following the bio-psycho-social model. For an appropriate clinical practice during the onabotulinumtoxinA prophylactic treatment, we recommend an integrated approach, including clinical psychologists working in collaboration with the Neurorehabilitation professionals, to assess psychological needs and profiles. The observed association between coping strategies, CM disability and pharmacological management highlights the importance of introducing and tailoring psychological interventions considering different psychological factors, including coping strategies. Therefore, the current findings, although being explorative, reinforce the need for using a multidisciplinary approach to implement tailored brief psychological interventions aiming at reducing psychological distress and improving functional coping strategies and self-efficacy in the CM management. As regards the psychological treatment in the neurorehabilitation setting, different approaches, such as cognitive-behavioral, mindfulness, relaxation and biofeedback approaches, have been studied and reviewed resulting valid and safe interventions; and they should be introduced and used in the management of patients with headache [45]. Although these approaches vary somewhat, they have all shown to help modifying maladaptive behaviors and ways of thinking that contribute to headache-related burden and pain [43, 44].”

In the conclusion section, we explain that the applicability of our observations in clinical practice requires a multidisciplinary context with the strict collaboration of specialist in neurology, clinical psychology and neurorehabilitation.

As to the applicability of our findings, we stated that the psychological assessment would be more applicable in a multidisciplinary context such as the neurorehabilitation setting.

The paper will benefit from a proof-read for the language edits.

Authors reply

The language has been reviewed to correct any mistake and typos.

Reviewer 2 Report

In their introduction the authors should include some references (and short discussion) of newer medications which also seem to be indicated for the prophylaxis of chronic migraine. 

Is the administration of botulin toxin every 12 weeks in accordance with standard of care (according to international guidelines)? The authors seem to indicate in their discussion that no standard application protocol was applied for the administration of the drug - this is a major short-coming of the study since previous publications have clearly shown the necessity of a very standardized application of botulin toxine.

Why not re-administration upon reappearance of migraine symptoms?

Were the performed post-hoc testing included and described in the original research protocol befor inclusion of first patient?

Limitation is of course that psychological testing was not repeated after first (or couple) of administrations to investigate the impact of the treatment on the overall status of the patient. This limiots the merit of the performed study. In the future the effect of botulin alone on the status of the patient should be examined but also compared to an approach combining botuline and cognitive-behavioral therapy.

Author Response

Manuscript ID: toxins-576518

Title: A prospective observational cohort study on pharmacological habitus, headache-related disability and psychological profile in patients with chronic migraine undergoing OnabotulinumtoxinA prophylactic treatment.

Dear Editor,

I wish to thank you for your assistance with the manuscript entitled “A prospective observational cohort study on pharmacological habitus, headache-related disability and psychological profile in patients with chronic migraine undergoing OnabotulinumtoxinA prophylactic treatment.”

The manuscript has been revised according to all the Reviewers concerns. The introduction and discussion have been modified to reply to the Reviewers’ suggestions. The reference section has been updated accordingly. Tables have been improved in readability. The language has been reviewed to correct any mistake and typos.

Below a point-by-point reply to the Reviewers' comments is reported.

Thank you, once again, for considering Toxins as a potential home for our paper.

Yours sincerely,

Marialuisa Gandolfi, MD, PhD

Reviewers' comments:

Reviewer #2:

In their introduction the authors should include some references (and short discussion) of newer medications which also seem to be indicated for the prophylaxis of chronic migraine.

Authors reply

According to your suggestion, we included in the introduction a short paragraph on novel therapeutic options aimed at inhibiting calcitonin gene-related peptide signalling, which has attracted growing interest in the pharmacological research on migraine. Reference section has been updated, accordingly [ref n.14-16].

Is the administration of botulin toxin every 12 weeks in accordance with standard of care (according to international guidelines)? The authors seem to indicate in their discussion that no standard application protocol was applied for the administration of the drug - this is a major short-coming of the study since previous publications have clearly shown the necessity of a very standardized application of botulin toxine. Why not re-administration upon reappearance of migraine symptoms?

Authors reply

Yes, it was. The administration of Botulinum toxin was every 12 weeks according to the international guidelines based on the PREEMPT injection protocol. A total of 195 U was injected every 12 weeks into 39 injection sites (each injection was 5U/0.1 mL). The PREEMPT injection protocol should be followed, since it is the only protocol that has proved efficacy of onabotulinumtoxinA, as recommended by the Guidelines recently published [Bendtsen L, Sacco S, Ashina M, Mitsikostas D, Ahmed F, Pozo-Rosich P, Martelletti P. Guideline on the use of onabotulinumtoxinA in chronic migraine: a consensus statement from the European Headache Federation. J Headache Pain. 2018 Sep 26;19(1):91]. We clarified in the first paragraph of the discussion that the same specialist performed the BoNT/A injection according to the international guidelines protocol. Following the international guidelines for the BoNT/A injection, which is based on the PREEMPT injection protocol, the prophylactic treatment was administered every 12 weeks. We clarified this point in the methods section.

Were they performed post-hoc testing included and described in the original research protocol before inclusion of first patient?

Authors reply

The present project is part of the comprehensive EXPLORE project, as reported in the methods section. In the original Explore protocol, both clinical and psychological outcomes were described according to the aim of the project. For the specific aims of the present study, we followed the Explore protocol, and we also included routine clinical data collected using the headache diary, which is routinely filled by patients. Post-hoc testing has been added for the specific aims of the present study.

Limitation is of course that psychological testing was not repeated after first (or couple) of administrations to investigate the impact of the treatment on the overall status of the patient. This limits the merit of the performed study. In the future the effect of botulin alone on the status of the patient should be examined but also compared to an approach combining botuline and cognitive-behavioral therapy.

Authors reply

According to your concern we included in the discussion that this longitudinal study should be introduced as a future direction of our study in order to explore the effect of onabotulinumtoxinA on psychopathology course both alone and combining botulinum and effective psychological approaches. Future studies should investigate the impact of the prophylactic treatment (OnabotulinumtoxinA or monoclonal antibodies) on the overall status of the patient, to repeat psychological assessment after the first three onabotulinumtoxinA administration treatment cycles and to examine the combined effects of prophylactic treatment and psychological approaches (e.g. cognitive-behavioural therapy). The reference section has been updated, accordingly.
